# Psycho-Electrophysiological Benefits of Forest Therapies Focused on Qigong and Walking with Elderly Individuals

**DOI:** 10.3390/ijerph18063004

**Published:** 2021-03-15

**Authors:** Jiyune Yi, Seul Gee Kim, Taegyu Khil, Minja Shin, Jin-Hee You, Sookja Jeon, Gue Hong Park, Ah Young Jeong, Youngsuwn Lim, Kahye Kim, Jingun Kim, Byunghoon Kang, Jueun Lee, Jeong Hwan Park, Boncho Ku, Jungmi Choi, Wonseok Cha, Hwa-Jin Lee, Changseob Shin, Wonsop Shin, Jaeuk U. Kim

**Affiliations:** 1Department of Forest Therapy, Chungbuk National University, Cheonngju, Chungbuk 28644, Korea; jiyuneyi@gmail.com (J.Y.); ktg0704@hanmail.net (T.K.); yeamolove@hanmail.net (M.S.); volustas@hanmail.net (J.-H.Y.); dawon0619@hanmail.net (S.J.); hoee21@hanmail.net (G.H.P.); honggilyue@naver.com (A.Y.J.); suwnmail@naver.com (Y.L.); jingun0308@naver.com (J.K.); byunghoon21@naver.com (B.K.); yjueun32@hanmail.net (J.L.); sinna@chungbuk.ac.kr (C.S.); 2Future Medicine Division, Korea Institute of Oriental Medicine, Daejeon 34054, Korea; sgkim11@kiom.re.kr (S.G.K.); kkh2@kiom.re.kr (K.K.); siegfriegd@kiom.re.kr (J.H.P.); secondmoon@kiom.re.kr (B.K.); 3Human Anti-Aging Standards Research Institute, Uiryeong, Gyungnam 52151, Korea; jmchoi@brnd.co.kr (J.C.); danho@brnd.co.kr (W.C.); 4Acupuncture & Meridian Science Research Center, Kyung Hee University, Seoul 02447, Korea; hwajin_lee@khu.ac.kr

**Keywords:** forest therapy, cognitive impairment, dementia, Qigong, walking in the forest, psychology, electrophysiology, electroencephalography, heart rate variability, bioimpedance

## Abstract

We developed two distinct forest therapy programs (FTPs) and compared their effects on dementia prevention and related health problems for older adults. One was focused on Qigong practice in the forest (QP) and the other involved active walking in the forest (WP). Both FTPs consisted of twelve 2-h sessions over six weeks and were conducted in an urban forest. We obtained data from 25, 18, and 26 participants aged 65 years or above for the QP, WP, and control groups, respectively. Neuropsychological scores via cognition (MoCA), geriatric depression (GDS) and quality of life (EQ-5D), and electrophysiological variables (electroencephalography, bioimpedance, and heart rate variability) were measured. We analyzed the intervention effects with a generalized linear model. Compared to the control group, the WP group showed benefits in terms of neurocognition (increases in the MoCA score, and alpha and beta band power values in the electroencephalogram), sympathetic nervous activity, and bioimpedance in the lower body. On the other hand, the QP group showed alleviated depression and an increased bioimpedance phase angle in the upper body. In conclusion, both active walking and Qigong in the forest were shown to have distinctive neuropsychological and electrophysiological benefits, and both had beneficial effects in terms of preventing dementia and relieving related health problems for elderly individuals.

## 1. Introduction

Population ageing is a global phenomenon; there were 703 million persons aged 65 years or older in the world in 2019, and it is estimated that the number of older persons will double by 2050. In 2015−2020, a person aged 65 years was expected to live an additional 17 years, which could increase to 19 years by 2045−2050 [1]. In a rapidly ageing population, chronic health problems have replaced infectious diseases as the dominant health care burden. The prevalence of multimorbidity in older persons ranges between 55% and 98%, and results in disability and functional decline, a poor quality of life, and high healthcare costs for the older population [2].

For the prevention and regulation of age-related chronic health problems, activities in nature and forest bathing are gaining attention as lifestyle-related interventions [3]. More recently, the concept of forest therapy has been proposed as a more active intervention, adding guided practice that could engage participants in various therapeutic activities in the forest for restoration, rehabilitation, and wellness [4,5,6,7]. Forest therapy is relatively new concept in many countries [8], but is becoming popular as an active healthcare methodology. In particular, in South Korea, by law, forest therapy was recently defined as “immune-strengthening and health-promoting activities, which utilize a variety of elements of the forest” [9]. Forest therapy has been reported to have positive benefits on physical and mental health and social wellbeing [10,11]. The psychological benefits include effects on depression, anxiety, mood, mental relaxation, attention restoration, and subjective pain. The physiological benefits include effects on the immune system (e.g., natural killer cells), insulin, oxidative stress, nervous system (e.g., heart rate), salivary cortisol, oxyhemoglobin, the cardio-circulatory system, and the respiratory system. In terms of social wellbeing, effects include social skills, emotional skills, self-esteem, and inspiration [4,12,13,14,15,16,17,18,19,20].

In the elderly population, neurodegenerative diseases, especially dementia, have become a leading cause of death and a major socioeconomic cost. According to the World Health Organization, there are around 50 million people with dementia, with 10 million new cases diagnosed every year, and the number of cases is predicted to increase to 152 million by 2050. The global cost of care for patients with dementia was estimated to be 818 billion USD, and 85% of costs are related to family and social care [21]. To date, no medication has been identified as an effective cure of dementia, and recent research estimated that approximately one-third of dementia cases could be prevented by modifying risk factors [22]. Risk factors of dementia include hypertension, alcohol abuse, obesity, smoking, depression, social isolation, physical inactivity, diabetes, and air pollution [22]. Many of these factors can be appropriately regulated and promoted by lifestyle modifications and interventions such as physical exercise, cognitive interventions, social activities, and activities in nature [23].

Some forest therapies conducted for the elderly population are based on or run in combination with walking, physical exercise, cognitive intervention, and social activity [24,25,26]. The reported benefits include effects on blood pressure, arterial stiffness, pulmonary function, salivary cortisol level, quality of life, depression and anxiety, mood state, and blood biomarkers related to cardiovascular disease [6,25,27,28,29]. Relieving these symptoms could be beneficial for dementia patients. Therefore, in aged and super-aged societies, forest therapy could be used as a major modality of preventive intervention for dementia. Despite its potential, the use of forest therapy for older populations at risk of dementia has only become of interest recently, and evidence for its benefit is still low because of the small numbers of participants, preliminary study designs, or short program hours used in previous research [24,30]. It is essential to develop forest therapy programs (FTPs) that are effective for dementia prevention.

Senile neurocognitive decline is a slow process that occurs over the course of years [31], and older people have low mobility and degraded physical and cognitive abilities. Therefore, undergoing adequate FTP to prevent dementia should be done long term, should present a low burden in terms of physical and cognitive resources, and should involve a forest environment with easy access. For this reason, recent FTPs with older Korean populations have mostly been conducted in urban forests [24,30,32]. In particular, our previous work compared two FTPs, one focused on forest walking and the other on guided meditative breathing [24]. The FTP involving meditative breathing was purposefully tested for its health promotion and dementia prevention effects in elderly people, especially those with low mobility. We reported that both FTPs were effective at improving electrophysiological markers, as shown by electrophysiology (EEG), heart-rate variability (HRV), and bioimpedance assessments. The program ran for 11 weeks with one 2-h session per week.

From our previous work, we found that the FTP focused on guided meditative breathing was more beneficial as it encouraged focused attention and rhythmic movements. Qigong practice can meet such needs as it integrates deep-breathing techniques, meditation, focused attention, and rhythmic moving or stationary body postures [33]. Qigong can be categorized into medical or martial arts types. Medical Qigong is defined as “the coordination of gentle exercise and relaxation through meditation and breathing exercises based on the Chinese medicine theory of energy channels” [34]. Recent systematic reviews and meta-analyses show that Qigong benefits the general quality of life, reduces hypertension, aids in fall prevention and balance, improves cognitive performance and muscle strength, reduces symptoms associated with osteoarthritis and chronic obstructive pulmonary disease, and reduces pain. In addition, it has psychological benefits such as reductions in anxiety and depression and improved self-confidence [35,36,37]. Qigong can be practiced anywhere by almost anyone, irrespective of age or physical condition. Therefore, FTP incorporating appropriate Qigong practice would have synergetic effects on health, including dementia prevention; yet, little is known about the beneficial effects of Qigong as a FTP.

In this study, we aim to develop FTPs that can be run in urban forest areas for the elderly population to prevent cognitive decline. Specifically, we propose two FTPs: a Qigong-based program and a forest walking program. The two FTPs are hypothesized to induce physiological benefits differently through different underlying mechanisms. We compare the effects of the FTPs on neuropsychological and electrophysiological variables, as potential risk factors of dementia. We hope that these variables can be used as biomarkers for forest therapies in the future, as electrophysiological assessments provide objective outcomes and are portable, noninvasive, and cost-effective.

## 2. Materials and Methods

### 2.1. Forest Therapy Program

By reviewing the results of previous FTPs studied by us as well as the body of literature focused on promoting the health status of the elderly [35,36,38], we developed two FTPs with modifications compared with previous approaches [24]. Each FTP was composed of 12 sessions, with two sessions per week (six weeks in total) and two hours per session (10:00 a.m.–12:00 p.m.). The FTPs were conducted in an urban forest located in Cheongju City, Korea (M-forest), which was easily reachable for nearby urban residents.

To maximize the effectiveness of the interventions, both FTPs incorporated therapeutic components of Korean medicine. One program was called the “Qigong program (QP)” and the other program was called the “walking program (WP)”. The QP consisted of 10 min of warm up exercises (body tapping), 50 min of guided Qigong exercises, 20 min of band gymnastics (arm stretching and shoulder lifting), 30 min of physio-cognitive play (various types of clapping, artwork with leaves and natural materials), and 10 min of cool down exercises (light stroll). Likewise, WP consisted of 10 min of warm up (body tapping), 30 min of physio-cognitive play (various types of clapping, line dance), 20 min of band gymnastics (arm stretching and shoulder lifting), 50 min of active walking, and 10 min of cool down activities (oil massage).

The key component in QP was guided Qigong exercise, which is composed of meridian relaxation and body scanning, body balancing and sensory attention, motion-coherent breathing techniques, and Taoin massage [38]. It aims to increases the sensory awareness of the body, activate therapeutic Qi and blood circulation through the meridian system and peripheral arteries, and stabilize the Qi (a cool down process) [38]. On the other hand, the key component of WP was active walking in the forest. Yongquan (K11) acupoints were stimulated while walking by taping red beans to the Yongquan points. This is known to have beneficial effects on cognition, hypertension and the blood circulation, and sleep disorders, among others [39,40]. In the body tapping session, participants tapped their whole bodies, and several acupoints known to be associated with dementia prevention, such as Baihui (GV20), Shenting (GV24), Fengchi (GB20), and Taixi (KI3), were used [41]. The difficulty level of each program increased every three sessions as the participants became accustomed to each level. In particular, Qigong exercise and active walking sessions aimed at the level of 12 to 13 points on the Borg rating of perceived exercise for the first to third sessions, and this increased to 16−17 points for the last three sessions [42].

### 2.2. Subject and Study Protocol

The FTPs ran from May to June, 2019. We aimed to recruit 30 participants for each FTP and for the control group (CN). The participants in the CN received no intervention or treatment related to activities in the forest. The inclusion criteria were as follows:-Aged 65 years or older;-Not diagnosed with dementia;-No restrictions on outdoor activity for more than 3 h;-Able to communicate and complete the self-report questionnaires;-Able to understand the purpose of the study and voluntarily sign the consent form.

At the time of recruitment, we examined the general health status of the participants through several questionnaires. First, we acquired basic demographic information that could affect cognitive and physical health, such as sex, age, height, weight, smoking, alcohol, education level, and medical history. Second, to exclude subjects with suspected dementia or functional disabilities due to neurodegenerative diseases, we used the MMSE-DS [43], the Korean version of the Mini Mental-Status Examination (MMSE) to assess global cognitive status, and the Korean instrumental activities of daily living (K-IADL) [44].

Subjects were recruited through advertisements and phone-calls, and we were assisted by one health center, one senior club, and an ecological cultural center. Written informed consent was obtained from each subject at enrollment. The study was approved by the Institutional Review Board of Chungbuk National University (IRB number: CBNU-201903-BMSB-812-01).

### 2.3. Test Instrument and Measurement Protocol

The effects of FTPs were tested to identify any neuropsychological (cognition, depression, quality of life) and electrophysiological (neurophysiology by electroencephalography (EEG), body composition and cellular metabolic states by bioimpedance, and autonomic nerve response by heart rate variability (HRV)) changes before and after each FTP session. For the cognition testing, we used the Korean version of the Montreal Cognitive Assessment (MoCA), which is widely used to detect mild cognitive impairment in the elderly population [45,46]. For the depression test, we used the Korean version of the Geriatric Depression Scale (GDS), which is suitable for evaluating the depression levels of elderly people [47,48]. To test the participants’ general quality of life, we used the EQ-5D, which was developed by the EuroQol group [49].

In terms of electrophysiological factors, we measured EEG signals to evaluate neurocognitive function [50,51], bioimpedance to evaluate the body composition and cellular metabolic states [52], and photoplethysmography (PPG) signals to evaluate the autonomic nerve response based on an HRV analysis [53]. Numerous studies have demonstrated that patients with Alzheimer’s disease (AD) have increased delta and theta band powers and decreased alpha and beta band powers with frequency lowering of the power density peak [54]. Therefore, some key EEG variables, such as MEF, Pα, Pβ, and ATR (listed in Table A1) are expected to decrease with cognitive decline. On the other hand, bioimpedance analysis (BIA) measures impedance data, such as impedance, reactance, and the phase angle, and offers information on health-related quantities such as cellular health (intracellular water, extracellular water, total body water, etc.) and body composition (fat mass, fat-free mass (FFM), protein mass, mineral mass, etc.) (see Table A1) [52]. A promising application of BIA is the estimation of metabolic malnutritional states that can occur due to ageing [55], metabolic syndrome [56], or cancer [57]. Decreases in the phase angle (PhA) and FFM are sensitive indicators of the metabolic malnutritional state. Lastly, HRV is a measure of variability between adjacent heart beats and is commonly used to estimate the autonomic nervous response [53]. For instance, the spectral power of the high frequency component (HF) represents parasympathetic modulation and that of the low frequency component (LF) relative to the HF or HF+LF represents sympathetic modulation [53] (see Table A1).

For the EEG measurement, a wireless device (neuroNicle FX2, LAXTHA, Inc., Daejeon, Korea) was used. This device measured EEG signals in the prefrontal region of Fp1 and Fp2 according to the International 10/20 electrode system. The device’s sampling rate was 250 Hz, the bandpass frequency was 3 to 43 Hz, and all contact impedances were kept below 10 kΩ. The EEG signals were recorded for five minutes after signals were stabilized with the subject seated in a resting state with their eyes closed and muscles relaxed in a quiet room. A trained operator monitored the subject and EEG signals and alerted the subject whenever s/he showed signs of behavioral artifacts (eye rolling, blinking, muscle contractions, etc.) or drowsiness [58].

Bioimpedance was measured using the InBody S10 (InBody, Seoul, Korea). This measures raw data such as the impedance and reactance at multiple frequencies of 1, 5, 50, 250, and 1000 kHz with respect to five segments of the body (right/left arm, right/left leg, and trunk). The segmental PhA is then calculated at 50 kHz; PhA is defined as the angle between the impedance and reactance according to “PhA = arcsin (reactance/|impedance|) × 180°/π” [56]. For the measurement of segmental bioelectrical impedance, the subjects were assessed in a supine position with four electrodes in contact with the thumb and index finger of each hand, and another four electrodes were in contact with the interior and exterior sides of each ankle [56]. Health-related quantities such as the total body water or FFM can be derived from raw impedance data in combination with age, sex, weight, and height variables [52].

To measure HRV, a PPG device (ubpulse T1, LAXTHA, Inc., Daejeon, Korea) was placed on the fingertip of the left index finger of the subject for five minutes while they were in a comfortable, seated position. Ubpulse T1 has a sampling rate of 250 Hz and a bandpass frequency of 0.3 to 10.6 Hz. Four operators were adequately trained in the measurement of EEG, bioimpedance, and PPG. The electrophysiological measurements and some pictures of forest therapy sessions are shown in Figure 1.

### 2.4. Statistical Analysis

Statistical analyses were conducted using R software (ver. 1.3.1093) [59]. For the statistical test, the significance level was set to α = 0.05 (two-tailed). Missing values for the MMSE and demographic characteristics, such as education level, were imputed using the multiple imputation method provided by the “mice” package in R software [60]. Participants’ baseline characteristics for each FTP and the CN were summarized as the means and standard deviations (SDs) for continuous variables and the frequencies and proportions for categorical variables (Table 1). To investigate the differences in baseline characteristics, one-way analysis of variance (ANOVA) was used for continuous variables and the chi-squared test was used for categorical variables.

The main focus of this study was to compare changes in outcome variables within each FTP group and between each FTP group and the control group. The changes in outcomes at the endpoint (after completing each FTP) compared to at the baseline were analyzed using the generalized linear model (GLM), in which several confounders—age, sex, education level, daily activity hours, and MMSE score—were considered covariates (Table 2, Table 3, Table 4 and Table 5) [61]. The mean change and its 95% confidence interval (CI) within each FTP and the CN were presented according to each outcome variable. Multiple comparison tests were conducted between groups (i.e., QP vs. CN and WP vs. CN) with t-statistics to identify the mean difference in the change of each FTP from that of control group. *p*-values and 95% CIs calculated with the multiple tests were adjusted by Dunnett’s method [62]. The effect size of the mean change within each FTP intervention and the effect size of the mean difference in the change between each FTP and the control group presented in Table 2, Table 3, Table 4 and Table 5 were calculated using the method suggested by Rosnow et al. [63].

## 3. Results

A total of 90 subjects were recruited for the study from three recruitment sites. Of them, 16 were screened out due to the exclusion criteria, so 25 subjects remained in the QP group, 18 subjects in the WP group, and 26 in the CN group at the end of the study; no subject dropped out during the FTP sessions, while five subjects in the CN could not complete the post-test measurements due to private reasons. The dataset was screened prior to the analyses, and samples were excluded if they were missing demographic information, lost in the follow up period, featured extraction errors, or were missing data or pathophysiological features such as arrhythmia (Figure 2). As a result, 61 subjects remained in the questionnaire analysis, 59 in the EEG analysis, 66 in the bioimpedance analysis, and only 32 in the HRV analysis. As arrhythmia was a common health problem in our old-age participants, 37 subjects were excluded from the HRV analysis.

### 3.1. Demographics

Demographics and other variables collected from the study participants at baseline are summarized in Table 1. Regarding the major risk factors, we found no significant differences in age, MMSE score, marital status, daily activity hours, and most medical history items among the QP, WP, and CN groups; however, differences were found in sex and education level. Table 1 indicates that the level of bias due to discrepancies in demographic risk factors would have been marginal when interpreting the results.

### 3.2. Effects of Forest Therapy Programs

To test the psychophysiological changes mediated by the two FTPs, we performed an analysis of covariance (ANCOVA), with which we investigated changes in variables within each FTP (mean differences of measures between the session end and the baseline) and relative to individuals in the CN group. The statistical methods used are described in detail in Section 2.4. We adopted the GLM, where several confounders—age, sex, education level, MMSE, and daily activity hours—were considered covariates, and the mean differences in value of biomarkers compared with the control group were tested by t-statistics. The variables analyzed in each dataset are shown in Table A1. The test results are presented in the following subsections: Table 2, Section 3.2.1 shows changes in cognition, depression, and quality of life; Table 3, Section 3.2.2 shows the resting-state EEG response; Table 4, Section 3.2.3 shows the bioimpedance response; Table 5, Section 3.2.4 shows the HRV response.

#### 3.2.1. Cognition, Depression, and Quality of Life

As introduced in Section 2.3, we used MoCA, GDS, and EQ5D to assess cognition, depression, and quality of life, respectively. Table 2 shows the ANCOVA results. The MoCA score after each FTP was elevated in all groups compared with the baseline measures, as follows: CN ($\bar{\delta }$ = 2.09 with 95% CI of (0.76, 3.41), *p* < 0.01 and γ = 0.85), QP ($\bar{\delta }$ = 2.22 with 95% CI of (1.10, 3.34), *p* < 0.001 and γ = 0.89), and WP ($\bar{\delta }$ = 3.46 with 95% CI of (2.12, 4.80), *p* < 0.001 and γ = 1.34). A decrease in the GDS score was found in the QP group ($\bar{\delta }$ = −2.23 with 95% CI of (−4.49, 0.04), *p* < 0.1 and γ = 0.52), but no changes were observed in the CN and WP groups. Finally, the EQ-5D was increased in the WP group ($\bar{\delta }$ = 0.05 with 95% CI of (−0.00, 0.07), *p* < 0.1 and γ = 0.50), while no such increase was found in the QP and CN groups. In the multiple comparison analysis, no variable showed differences between groups in terms of *p*-values. In terms of the effect size, however, the WP group showed a marginal increase in the MoCA score with Γ = 0.46 ($\bar{\Delta }=1.37$) compared with the CN group, and the QP group showed a marginal decrease in GDS scores with Γ = 0.59 ($\bar{\Delta }=-2.63$).

#### 3.2.2. Resting-State EEG

The ANCOVA results of EEG variables are shown in Table 3. The variables analyzed were the median frequency (MEF), the power levels of the alpha and beta bands (Pα and Pβ), and the Pα/Pβ ratio (ATR) (details are in Table A1). With respect to the baseline values, no noticeable changes were found in the control group, but we found a decrease in the Pα ($\bar{\delta }$ = −0.28 with 95% CI of (−0.55, −0.00), *p* < 0.05 and γ = 0.45) and Pβ ($\bar{\delta }$ = −0.39 with 95% CI of (−0.71, −0.06), *p* < 0.05 and γ = 0.53) for the QP participants and an increase in the Pα ($\bar{\delta }$ = 0.33 with 95% CI of (−0.01, 0.67), *p* < 0.1 and γ = 0.54) for the WP participants. The multiple comparison analysis showed no differences between the FTP and CN groups in terms of *p*-values. In terms of the effect size, however, the QP showed a marginal increase in the MEF, with Γ = 0.43 ($\bar{\Delta }=0.22$), and a marginal decrease in the Pα, with Γ = −0.42 ($\bar{\Delta }=-0.29$) compared with the CN group, and the WP group showed marginal increases in Pα with Γ = 0.45 ($\bar{\Delta }=0.32$) and Pβ with Γ = 0.43 ($\bar{\Delta }=0.36$).

#### 3.2.3. Bioimpedance

Changes in bioimpedance variables following the FTP interventions are shown in Table 4. The investigated variables were fat-free mass (FFM), body fat mass (BFM), percent body fat (%BF = BFM/(FFM + BFM)), the impedances, reactances, and phase angles (PhAs) of the arms (Imp_arm, Reactance_arm and PhA_arm) and legs (Imp_leg, Reactance_leg and PhA_leg), and the phase angle of the whole body (PhA_body) (details are presented in Table A1). The FFM was reduced while the BFM did not change in the CN, QP, and WP groups. In the multiple comparison analysis, however, no significant differences were shown between the FTP intervention group and the CN group for both FFM and BFM.

Both impedance and reactance were reduced in the arms of individuals in the CN group, but they increased in the legs of QP and WP participants. PhA, which is the angle between reactance and resistance, decreased in the body and arms for individuals in all group, and increased in the legs for those in both the QP and WP groups. In the multiple comparison analysis, the impedance and reactance values of the arms and legs were shown to increase in both QP and WP participants compared to those in the CN. More importantly, the PhA of the arm (PhA_arm) increased in the QP group compared to the CN group ($\bar{\Delta }=0.14$ with 95% CI of (−0.03, 0.31) and Γ = 0.51), while no such increase was observed in the PhA of the leg (PhA_leg). In contrast, the PhA of the leg increased in the WP group compared to the CN group ($\bar{\Delta }=0.35$ with 95% CI of (0.10, 0.61), *p* < 0.01 and Γ= 0.86), while no such increase was observed in the PhA of the arm.

#### 3.2.4. Heart Rate Variability

HRV changes are shown in Table 5. The statistics tested were the spectral powers of high frequency (HF), low frequency (LF), percent of LF/(HF + LF) (%LF), total power (TP = HF + LF + VLF), and heart rate (HR) (details are presented in Table A1). Overall, no noticeable changes were found; for the LF, only a marginal decrease was shown for the WP group relative to the CN group ($\bar{\Delta }=-0.74$ with 95% CI of (−1.75, 0.27) and Γ = 0.44).

## 4. Discussion

In a previous study, we compared the electrophysiological benefits of two distinct FTPs for the elderly population. Each intervention was composed of 11 therapy sessions with one 2-h session per week [24]. In this work, we revised our previous approach to enhance the sustained effect of the FTPs. We investigated two FTPs composed of 12 sessions, with two sessions per week and two hours per session; we doubled the session frequency and halved the session period. For the WP group, we updated the previously developed walking program into 12 sessions; the main focus of the WP intervention was to make the participants walk actively to induce mild sweating. The previously used breathing program was substituted with a Qigong-centered program (QP), which was modified from a Qi Dance program originally developed for Parkinson’s disease so that it was suitable for the forest environment [38]. The QP was composed of three parts: (1) relaxing the meridians, (2) circulating Qi, and (3) stabilizing Qi. It was developed to be suitable for elderly people who have difficulty with active walking.

As a result, we measured bioimpedance behaviors similar to those presented in [24]. First, we did not observe significant changes in the FFM or BFM due to FTP intervention compared to the CN group. Second, we observed an increase in the phase angle of the legs (arms) in the WP (QP) group compared to that in the CN group, while no such changes were observed in the phase angles of other body parts (Table 4). This result confirms that the bioimpedance PhAs (at 50 kHz), independently of FFM or BFM and more sensitively than volumetric variables, can be used as a prognostic biomarker for the effects of an FTP. Indeed, our finding is in agreement with previous reports on metabolism and related diseases. For instance, an increase in PhA was observed following resistive training in older individuals [64], and decreases in PhA were observed in patients with diabetes mellitus [56], malnutrition [65], and chronic obstructive pulmonary disease [66], as well as those at risk of death in an intensive care unit [67] and cancer patients with a lower quality of life and malnutrition [57]. Therefore, bioimpedance PhAs (at 50 kHz) could be used as prognostic biomarkers of the FTP with respect to recovery from a metabolic malnutritional state [57,65]. The QP (WP) intervention was presumably effective at upregulating the metabolism of the upper (lower) body.

In contrast to the beneficial effect on the bioimpedance PhAs, the proposed FTPs were shown to have only minor benefits on neurophysiological markers, as assessed by EEG, and autonomic nerve responses, as measured by HRV. Specifically, compared to the CN group, the WP group showed increases in alpha and beta band powers (Pα and Pβ) with effect sizes of 0.43 and 0.45, respectively. However, the QP group showed an inconclusive result as it showed an increase in the median frequency (MEF) with an effect size of 0.43 but a decrease in the alpha band power (Pα) with an effect size of 0.42. As recent literature supports the proposal that the MEF and ATR decrease with cognitive decline [58], and alpha wave activity and beta wave power decrease in the early stage of Alzheimer’s disease [51,68,69], this implies that the WP intervention has possible neurophysiological benefits. In the HRV analysis, the WP group showed reduced LF power values, indicative of a reduced sympathetic response [53], with an effect size of 0.44, but no other benefits were observed.

Lastly, in terms of neuropsychological aspects, compared to the CN group, a cognitive benefit was observed in the WP group with an effect size of 0.46 for the MoCA score, and depression was observed to be alleviated in the QP group with an effect size of 0.59 for the GDS score. We note that in the WP group, moderate and consistent cognitive benefits were observed in terms of both the EEG markers and the MoCA score. There was no such consistent cognitive benefit in the QP group. However, the QP group showed a moderate benefit in terms of depression alleviation, consistent with previous reports [70,71].

Instantaneous and objective tools to assess the effects of forest therapy are needed for academic and commercial research purposes, and electrophysiology-based methods have high potential in this regard. HRV-based evaluation of the autonomic nervous system and cardiovascular function is commonly used in forest therapy [14,15,72,73], and portable EEG devices have recently been implemented to evaluate neurocognitive activities in urban green spaces [74,75]. More recently, we introduced a bioimpedance analysis (BIA) technique as an appropriate tool [24]; BIA offers reliable evaluation of body composition [76] and cellular metabolism [77].

This study has some limitations. First, the study period was rather short, and a longer period program should be attempted in the future. Since senile cognitive decline is a slow process occurring over years, exercise-based interventions for dementia prevention range from 6 to 52 weeks [78]. Next, there was no intervention for the control group. Through our survey, we found that the control group participants were actively involved in other activities such as gardening or singing schools, and the caloric consumption or equivalent activity hours of the replacement activities could not be followed up. Assuming the activity level of the control group ranged between equal to or less than that of the FTP participants, our organized FTPs were estimated to be more beneficial for selected electrophysiological and neuropsychological markers than equivalent level of similar outdoor/indoor activities (active control) or no activity (passive control). In a future study, the activity level of the control group should be appropriately controlled or evaluated to assess benefits of FTPs more rigorously. Lastly, more than 50% of the participants were excluded from HRV analysis, as many participants suffered from arrhythmias due to their old age.

Electrophysiological tools used in this study are low cost, portable, easy to implement to the field of forest therapy. They provide standardized scores which help evaluate objectively the effects of FTP intervention. It will induce participants’ active involvement in the program and provide increased credibility for their health benefits, which will eventually contribute to the expansion of forest therapy commercially as well as academically.

## 5. Conclusions

In conclusion, both the Qigong program and the walking program have health benefits for the elderly population in terms of effects on cognition, depression, and electrophysiology (such as bioimpedance, EEG and HRV). In particular, the Qigong program was found to be effective for alleviating depression and increasing the bioimpedance phase angle of the upper body, and the walking program was effective for preventing cognitive decline and increasing the phase angle of the lower body. This suggests that both active walking and Qigong practice in the forest, in combination with other forest activities, can be beneficial for dementia prevention. Based on these results, in a future study, we plan to upgrade the forest therapy program to include more sessions and a longer timeframe as well as including an appropriate control or quantification method to assess the activity level of the control group. We hope to examine the effects related to the reduction in more diverse risk factors for dementia.

## Figures and Tables

**Figure 1 ijerph-18-03004-f001:**
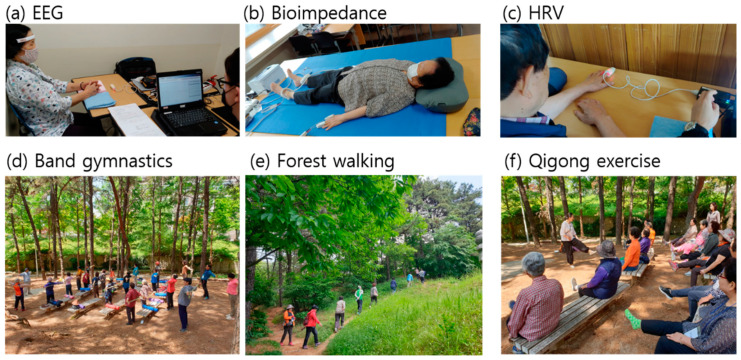
Measurement of (**a**) EEG, (**b**) bioimpedance, and (**c**) HRV, and pictures of forest therapy sessions involving (**d**) band gymnastics, (**e**) forest walking, and (**f**) Qigong exercise.

**Figure 2 ijerph-18-03004-f002:**
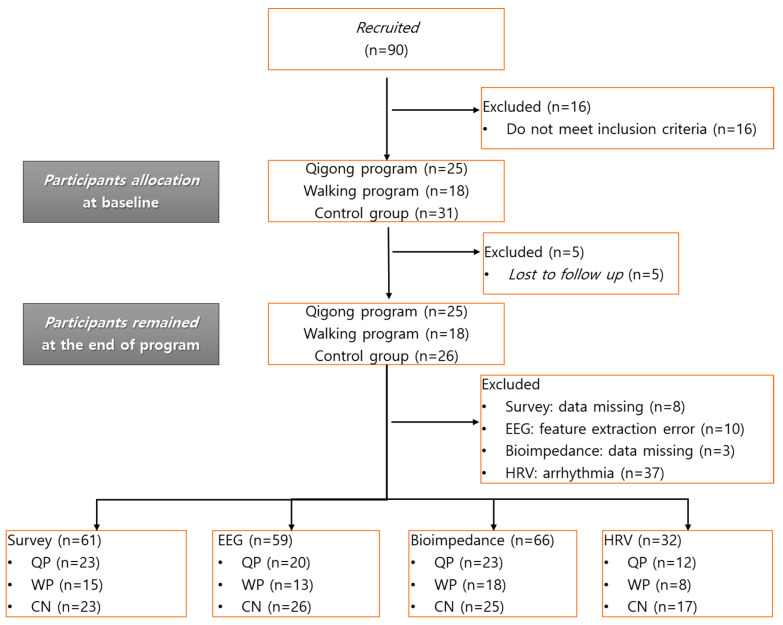
Study flow chart.

**Table 1 ijerph-18-03004-t001:** Demographic information.

	Control	Qigong Program	Walking Program	*p*-Value
*N* (%)	31 (41.9%)	25 (33.8%)	18 (24.3%)	
Sex				0.003
Male	21 (67.7%)	6 (24.0%)	11 (61.1%)	
Female	10 (32.3%)	19 (76.0%)	7 (38.9%)	
Age (y)	75.7 ± 4.2	75.1 ± 4.9	78.0 ± 5.5	0.145
Height (cm)	160.0 ± 10.1	154.2 ± 7.3	159.1 ± 8.4	0.047
Weight (kg)	63.6 ± 11.8	60.7 ± 7.9	62.3 ± 8.3	0.558
BMI (kg/m^2^)	24.7 ± 3.4	25.6 ± 3.5	24.5 ± 2.2	0.490
MMSE	25.7 ± 2.4	26.8 ± 2.0	26.2 ± 2.2	0.170
Smoking: Yes	1 (3.3%)	2 (9.5%)	0 (0.0%)	0.338
Alcohol: Yes	10 (33.3%)	8 (40.0%)	3 (17.6%)	0.327
Religion: Yes	22 (73.3%)	17 (68.0%)	10 (55.6%)	0.444
Marital status: Married	17 (56.7%)	6 (24.0%)	8 (44.4%)	0.050
Education level	6.2 ± 4.0	5.2 ± 4.2	8.4 ± 2.5	0.023
Medical History				
Hypertension: Yes	9 (42.9%)	13 (59.1%)	8 (61.5%)	0.456
Diabetes: Yes	8 (38.1%)	6 (27.3%)	3 (23.1%)	0.601
Dyslipidemia: Yes	3 (14.3%)	12 (54.5%)	3 (23.1%)	0.013
Arthritis: Yes	6 (28.6%)	9 (40.9%)	3 (23.1%)	0.500
Cerebrovascular disease: Yes	2 (9.5%)	1 (4.5%)	1 (7.7%)	0.815
Depression: Yes	1 (4.8%)	0 (0.0%)	0 (0.0%)	0.428
Other	10 (50.0%)	11 (50.0%)	5 (38.5%)	0.767
Daily activity hours (h/day)	1.0 ± 0.8	1.0 ± 0.9	1.1 ± 0.9	0.880

Data are presented as the means ± SDs for continuous variables and as the frequencies and proportions for categorical variables. *p*-values are obtained from a one-way ANOVA test for continuous variables and a chi-squared test for categorical variables. BMI: Body Mass Index; MMSE: Mini Mental-Status Examination.

**Table 2 ijerph-18-03004-t002:** Changes in cognition score (MoCA), depression level (GDS), activity level of daily living (IADL), and quality of life (EQ-5D) due to the forest therapy program (FTP) intervention (ANCOVA results).

	Control			Qigong Program			Walking Program			QP–CN		WP–CN	
	${\bar{X}}_{B}$	$\bar{\delta }$ (95% CI)	γ	${\bar{X}}_{B}$	$\bar{\delta }$ (95% CI)	γ	${\bar{X}}_{B}$	$\bar{\delta }$ (95% CI)	γ	$\bar{\Delta }$ (95% CI)	Γ	$\bar{\Delta }$ (95% CI)	Γ
MoCA	19.18	**2.09 **** **(0.76, 3.41)**	**0.85**	20.48	**2.22 ***** **(1.10, 3.34)**	**0.89**	20.67	**3.46 ***** **(2.12, 4.80)**	**1.34**	0.14 (−1.88, 2.15)	0.05	1.37 (−0.82, 3.56)	**0.46**
GDS	12.45	0.40 (−2.28, 3.09)	0.10	14.33	**−2.23·** **(−4.49, 0.04)**	**0.52**	7.07	−0.03 (−2.73, 2.66)	0.01	−2.63 (−6.64, 1.38)	**0.59**	−0.44 (−5.02, 4.14)	0.09
EQ-5D	0.84	0.01 (−0.05, 0.07)	0.09	0.84	−0.00 (−0.05, 0.04)	0.02	0.85	**0.05·** **(−0.00, 0.11)**	**0.50**	−0.01 (−0.10, 0.07)	0.11	0.04 (−0.05, 0.14)	0.35

The changes in variables between the end and baseline periods of the study were analyzed using a GLM. ${\bar{X}}_{B}$ represents the mean value at baseline, $\bar{\delta }$ (95% CI) and γ are the mean (95% confidence interval) and effect size of the difference between the end and baseline periods of each program. Multiple comparisons were conducted to identify the mean difference in changes occurring in individuals participating in each FTP compared with the CN group with t-statistics, where $\bar{\Delta }$ and Γ represent the mean difference and effect size of individuals in each FTP relative to those in the CN group. *p*-values (**·**
*p* < 0.1, * *p* < 0.05, ** *p* < 0.01, *** *p* < 0.001) and 95% CIs were adjusted by Dunnett’s method, and effect sizes were calculated by the Rosnow method. Presented in boldface for the mean difference with *p* < 0.1, and for the effect size > 0.4.

**Table 3 ijerph-18-03004-t003:** Changes in EEG variables following the FTP interventions (ANCOVA results).

	Control			Qigong Program			Walking Program			QP−CN		WP−CN	
	${\bar{X}}_{B}$	$\bar{\delta }$ (95% CI)	γ	${\bar{X}}_{B}$	$\bar{\delta }$ (95% CI)	γ	${\bar{X}}_{B}$	$\bar{\delta }$ (95% CI)	γ	$\bar{\Delta }$ (95% CI)	Γ	$\bar{\Delta }$ (95% CI)	Γ
MEF (Hz)	9.06	−0.13 (−0.31, 0.06)	0.27	9.16	0.09 (−0.11, 0.29)	0.20	9.03	0.04 (−0.21, 0.29)	0.10	0.22 (−0.11, 0.54)	**0.43**	0.17 (−0.19, 0.53)	0.32
$P\alpha \;\left[{\mu V}^{2}\right]$	2.44	0.01 (−0.24, 0.26)	0.02	2.59	**−0.28 *** **(−0.55, −0.00)**	**0.45**	2.52	**0.33·** **(−0.01, 0.67)**	**0.54**	−0.29 (−0.73, 0.15)	**0.42**	0.32 (−0.17, 0.81)	**0.45**
$P\beta \;\left[{\mu V}^{2}\right]$	2.99	−0.17 (−0.46, 0.12)	0.24	3.21	**−0.39 *** **(−0.71, −0.06)**	**0.53**	3.03	0.19 (−0.21, 0.58)	0.26	−0.21 (−0.73, 0.30)	0.27	0.36 (−0.21, 0.93)	**0.43**
ATR	1.12	0.08 (−0.02, 0.17)	0.33	1.12	0.08 (−0.03, 0.18)	0.34	1.15	0.04 (−0.09, 0.16)	0.16	0.00 (−0.16, 0.17)	0.01	−0.04 (−0.22, 0.14)	0.15

The methodological details are identical to those presented in the footer of Table 2 and definition of variables are presented in Table A1.

**Table 4 ijerph-18-03004-t004:** Changes in bioimpedance variables following the FTP interventions (ANCOVA results).

	Control			Qigong Program			Walking Program			BP−CN		WP−CN	
	${\bar{X}}_{B}$	$\bar{\delta }$ (95% CI)	γ	${\bar{X}}_{B}$	$\bar{\delta }$ (95% CI)	γ	${\bar{X}}_{B}$	$\bar{\delta }$ (95% CI)	γ	$\bar{\Delta }$ (95% CI)	Γ	$\bar{\Delta }$ (95% CI)	Γ
FFM (kg)	47.27	**−0.57·** **(−1.20, 0.06)**	0.36	40.80	**−0.68·** **(−1.37, 0.00)**	**0.42**	44.65	**−1.05 **** **(−1.80, −0.29)**	**0.66**	−0.12 (−1.22, 0.99)	0.06	−0.48 (−1.61, 0.64)	0.26
BFM (kg)	16.91	0.36 (−0.35, 1.07)	0.20	20.24	0.63 (−0.14, 1.40)	0.34	17.62	0.47 (−0.37, 1.31)	0.26	0.27 (−0.98, 1.52)	0.13	0.11 (−1.14, 1.37)	0.05
%BF (%)	26.33	0.58 (−0.31, 1.47)	0.26	32.67	**1.08 *** **(0.11, 2.05)**	**0.46**	28.20	**1.07 *** **(0.02, 2.13)**	**0.48**	0.50 (−1.08, 2.08)	0.19	0.49 (−1.09, 2.08)	0.19
Imp_arm (Ω)	312.72	**−7.13 *** **(−12.62, −1.63)**	**0.52**	333.27	2.08 (−3.92, 8.07)	0.14	325.74	0.89 (−5.79, 7.58)	0.06	**9.20·** **(−0.42, 18.82)**	**0.58**	8.02 (−1.94, 17.98)	**0.49**
Imp_leg (Ω)	164.66	−0.99 (−6.41, 4.44)	0.07	159.62	**15.63 ***** **(9.97, 21.29)**	**1.15**	150.42	**13.06 ***** **(6.89, 19.22)**	**1.00**	**16.62 ***** **(7.05, 26.18)**	**1.05**	**14.04 **** **(4.40, 23.69)**	**0.89**
Reactance_arm (Ω)	31.67	**−2.68 ***** **(−3.66, −1.70)**	**1.09**	30.77	−0.86 (−1.91, 0.20)	0.34	32.09	**−2.09 ***** **(−3.27, −0.91)**	**0.84**	**1.82 *** **(0.12, 3.53)**	**0.65**	0.59 (−1.18, 2.36)	0.20
Reactance_leg (Ω)	14.91	−0.14 (−0.86, 0.57)	0.08	13.85	**1.84 ***** **(1.06, 2.63)**	**0.98**	12.73	**2.15 ***** **(1.29, 3.02)**	**1.17**	**1.99 **** **(0.72, 3.25)**	**0.95**	**2.30 ***** **(1.01, 3.58)**	**1.10**
PhA_body	5.62	**−0.22 ***** **(−0.32, −0.12)**	**0.89**	5.28	**−0.15 **** **(−0.25, −0.04)**	**0.56**	5.45	−0.08 (−0.20, 0.04)	0.32	0.07 (−0.10, 0.25)	0.26	0.14 (−0.03, 0.32)	**0.50**
PhA_arm	5.86	**−0.38 ***** **(−0.47, −0.28)**	**1.58**	5.33	**−0.23 ***** **(−0.34, −0.13)**	**0.94**	5.65	**−0.32 ***** **(−0.43, −0.21)**	**1.35**	0.14 (−0.03, 0.31)	**0.51**	0.05 (−0.11, 0.22)	0.20
PhA_leg	5.23	0.02 (−0.13, 0.16)	0.04	4.98	0.12 (−0.03, 0.28)	0.33	4.88	**0.37 ***** **(0.20, 0.54)**	**1.03**	0.11 (−0.15, 0.37)	0.26	**0.35 **** **(0.10, 0.61)**	**0.86**

The methodological details are identical to those presented in the footer of Table 2 and definition of variables are presented in Table A1.

**Table 5 ijerph-18-03004-t005:** Changes in HRV variables following the FTP interventions (ANCOVA results).

	Control			Qigong Program			Walking Program			BP−CN		WP−CN	
	${\bar{X}}_{B}$	$\bar{\delta }$ (95% CI)	γ	${\bar{X}}_{B}$	$\bar{\delta }$ (95% CI)	γ	${\bar{X}}_{B}$	$\bar{\delta }$ (95% CI)	γ	$\bar{\Delta }$ (95% CI)	Γ	$\bar{\Delta }$ (95% CI)	Γ
HF (msec^2^)	4.62	0.19 (−0.28, 0.66)	0.16	4.85	−0.11 (−0.61, 0.40)	0.09	4.88	−0.28 (−0.85, 0.30)	0.23	−0.29 (−1.11, 0.52)	0.21	−0.46 (−1.32, 0.39)	0.33
LF (msec^2^)	4.48	0.40 (−0.16, 0.95)	0.28	4.51	−0.16 (−0.76, 0.43)	0.11	4.96	−0.34 (−1.02, 0.34)	0.24	−0.56 (−1.52, 0.41)	0.34	−0.74 (−1.75, 0.27)	**0.44**
%LF	49.36	1.25 (−0.97, 3.47)	0.22	47.99	−0.50 (−2.86, 1.86)	0.09	50.31	0.24 (−2.45, 2.94)	0.04	−1.75 (−5.58, 2.07)	0.27	−1.00 (−5.02, 3.01)	0.15
TP (msec^2^)	6.36	0.07 (−0.48, 0.62)	0.05	6.46	−0.25 (−0.83, 0.34)	0.17	6.54	−0.12 (−0.80, 0.55)	0.09	−0.32 (−1.27, 0.63)	0.20	−0.20 (−1.20, 0.80)	0.12
HR (bpm)	67.42	2.42 (−0.92, 5.76)	0.28	67.87	**2.97·** **(−0.58, 6.51)**	0.34	68.65	−0.44 (−4.49, 3.62)	0.05	0.54 (−5.21, 6.29)	0.06	−2.86 (−8.90, 3.18)	0.29

The methodological details are identical to those presented in the footer of Table 2 and definition of variables are presented in Table A1.

## Data Availability

The data presented in this study are available on request from the corresponding author.

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
