# Peer review of "Psycho-Electrophysiological Benefits of Forest Therapies Focused on Qigong and Walking with Elderly Individuals"

_ijerph, 2021, doi:10.3390/ijerph18063004_

Round 1

Reviewer 1 Report

Overall this is a well written paper with an interesting result on the area in this population.

The results are based on rational working hypothesis, well described and with a correct research design.

INTRODUCTION

The introduction provides sufficient background information for readers to understand the research aim, however the authors should clarify the importance of this topic and the actual knowledge in this area. It seems like some important clarification is missing.

Some of the sentences in introduction are too long and sometimes makes difficult its understanding.

Motivations for this study are more than clear and the objectives are clearly defined at the Introduction, the argumentation in this part was concise.

METHODS

The methodology proposed to reach the aim of the study look appropriate, well designed and conducted.

Specify the manufactured, company and country of all the instrument used in the research.

There are a few instances where assertions are made that are not substantiated with references.

RESULTS

Results paragraph include the most relevant data.

All of the tables must be reformat for better compression of the data.

DISCUSSION

All possible interpretations of the data considered are consistent, however, there are some grammar mistakes that should be corrected.

Conclusion respond to the research aim

Explain future research line according to the study conclusion and include practical application section

LITERATURE CITED

The literature cited is relevant to the study, but there are several instances in which the author makes assertions without substantiating them with references, but which are sustained by the main text and previous citations.

Reference style should be checked with the journal standards.

SIGNIFICANCE AND NOVELTY

As it stands, the results are novel and important enough for this journal.

Reviewer 2 Report

This paper is a follow-up study of previous works investigating Forest Therapy Programs for older individuals in South Korea, this time focusing on shorter-duration but more intensive programs of Qigong and Forest Walking therapies. The authors conclude that both Qigong and Forest Walking provide some neurophysiological and neuropsychological benefits, with Qingong more beneficial of depressive symptoms and upper body bioimpedance, and Forest Walking more beneficial for alleviating cognitive decline and lower body bioimpedance.

The results of the study, particularly the meticulous data, are of interest for future research on possible therapies for older individuals, and thus it is valuable to have this data available. The authors also present information about how therapies could be conducted and how to test their efficacy, which have implications for setting up similar studies in other countries facing similar challenges of aging populations such as Japan and Hong Kong as well as Europe.

There are a few aspects of the paper that could be improved. First, the data is presented in a very raw form, and there could be a bit more explanation about the implications of the data, how it relates to broader discussions of the needs of older individuals, and comparisons with other therapies.

Second, the lack of control in the "Control Group" is a bit unfortunate, and leads one to question how accurate the documented benefits of the two FTP really are. Considering that the Control Group was likely involved in many other activities that could be providing various neurophysiological and neuropsychological benefits, it makes it difficult to determine exactly what the specific effects of the FTP are. While this cannot be changed retroactively, the authors might be able to provide more information about why we their findings are not undermined by the lack of control over the Control Group. They might also offer some suggestions as to how to run future studies that better control for this issue.

Lastly, the paper requires some thorough English editing throughout.

If the authors are able to address these issues, then the paper could make a contribution to the field.

Round 2

Reviewer 2 Report

The authors have made good efforts to revise the paper, to better frame the contribution and limitations, and to better explain the data. I feel that the manuscript is now ready for publication. I congratulate the authors for their accomplishment.